# Sedentary Behaviour, Physical Activity and Life Satisfaction, Happiness and Perceived Health Status in University Students from 24 Countries

**DOI:** 10.3390/ijerph16122084

**Published:** 2019-06-13

**Authors:** Supa Pengpid, Karl Peltzer

**Affiliations:** 1ASEAN Institute for Health Development, Mahidol University, Salaya, Phutthamonthon, Nakhonpathom 73170, Thailand; supaprom@yahoo.com; 2Deputy Vice Chancellor Research and Innovation Office, North West University, Potchefstroom 2531, South Africa

**Keywords:** sedentary behaviour, physical activity, well-being, university students

## Abstract

The aim of this investigation was to estimate the independent and combined associations of sedentary behaviour (SB) and physical activity (PA) with life satisfaction, happiness and perceived health in university students. In a cross-sectional survey, 12,492 university students (median age 20 years, interquartile range = 3) from 24 countries responded to a questionnaire on SB, PA and well-being indicators. In adjusted linear regression, higher SB (4 to <8 h and ≥8 h) was associated with poorer life satisfaction (*β* = −0.21, confidence interval (CI): −0.27 to −0.14) and lower happiness (*β* = −0.31, CI: −0.46 to −0.17), and higher SB (≥8 h) was associated with lower perceived health (*β* = −0.08, CI: −0.13 to 0.03). In addition, moderate and/or high PA increased the odds for higher life satisfaction (*β* = 0.10, CI: 0.04 to 0.16), greater happiness (*β* = 0.27, CI: 0.15 to 0.39) and better perceived health (*β* = 0.12, CI: 0.08 to 0.15). Programmes that reduce SB and increase PA may promote life satisfaction, happiness and perceived health status in this university student population.

## 1. Introduction

In a systematic review, evidence was found for an association between sedentary behaviour (SB), physical inactivity and various forms of morbidity, including depression and anxiety, and also mortality [1,2]. Less is known about the relationship between SB and physical activity (PA) with life satisfaction, happiness and perceived health.

Few studies among different age groups, predominantly in countries with high income, found a negative association between SB and a positive association between PA and life satisfaction, happiness and perceived health. For example, among university students in Ireland, PA was associated with happiness and better perceived health [3]. Among university students in Croatia, PA was associated with life satisfaction [4]. Among adolescents in Iceland, lower SB and higher PA was associated greater life satisfaction [5], and among American college students, combined PA and SB was associated life satisfaction [6].

Among European adults from 15 countries, higher PA levels were in a positive dose-response relationship associated with greater happiness [7]. Among older adults in Norway, PA was associated with better perceived health [8]. In a longitudinal study, PA decreased the odds of unhappiness after two and four years [9]. In community-dwelling older adults, SB was associated with lower life satisfaction [10]. In a one-week intervention that induced SB, poorer life satisfaction was found [11]. Fewer studies have investigated the independent and combined associations of SB and PA with life satisfaction, happiness and perceived health in low- and middle-income countries. It would be important to investigate if associations between SB and PA with life satisfaction, happiness and perceived health can be found in a study across culturally different countries and regions in the world. Therefore, the aim of this investigation was to estimate the independent and combined associations of SB and PA with life satisfaction, happiness and perceived health in university students in 24 low- and middle-income countries.

## 2. Methods

### 2.1. Sample and Procedure

A cross-sectionally administered survey included 12,492 university students (median age 20 years, interquartile range = 3) with complete sedentary and physical activity measurements from 24 low- and middle-income countries (Bangladesh, Cameroon, Columbia, Grenada, India, Ivory Coast, Kenya, Jamaica, Kyrgyzstan, Laos, Madagascar, Malaysia, Mauritius, Myanmar, Namibia, Nigeria, Philippines, Russia, South Africa, Thailand, Tunisia, Turkey, Venezuela and Vietnam). Methodological procedures have been described previously [12]. Briefly, one university per country was selected by purposeful sampling. In each university, a stratified random sampling procedure was used to randomly select undergraduate students for a self-administered questionnaire survey and anthropometric measurements. Participating students signed informed consent forms, and all implementing institutions obtained ethics approvals: Ethics Review Committee North South University, Ethics Committee of the University of Yaoundé, Universidad de Pamplona Ethics Committee, St. George’s University Institutional Review Board, Ethics Committee of Institute of Technology and Institute of Sciences at GITAM (Gandhi Institute of Technology and Management) University, Félix Houphouët Boigny University Ethics Committee, University of Nairobi University of Nairobi Ethics and Research Committee, University of the West Indies Ethics Committee, Kyrgyz State Medical Academy Ethics Committee, The Ethics Committee of the University of Health Sciences, Ethics Committee of the University of Antananarivo, University of Malaya Medical Ethics committee (MECID 201412–905), University of Mauritius Research Ethics Committee, Research and Ethical Committee of University of Medicine 1, Research Ethics Committee of the University of Namibia, Ethics Review Committee Obafemi Awolowo University, Committee of the Western Visayas Health Research, Ethics Committee of the Peoples’ Friendship University of Russia, Medunsa Research and Ethics Committee (MREC/H/275/2012), Committee for Research Ethics (Social Sciences) of Mahidol University (MU-SSIRB 2015/ 116(B2), National Ethics Committee for Health Research at Institut National de la Santé Publique, Ethics Committee Istanbul University, Ethics Committee of the Universidad Central de Venezuela, and Committee of Research Ethics of Hanoi School of Public Health.

### 2.2. Measures

#### 2.2.1. Exposure Variables

*SB and PA* were assessed with the “International Physical Activity Questionnaire (IPAQ) short form” [13]. SB was classified into low (<4 h), moderate (4 to <8 h) and high (≥8 h) per day sedentary time [14], and PA was classified into three levels (low, moderate and high) of PA [15].

#### 2.2.2. Outcome Variables

Life satisfaction was assessed with the question, “All things considered, how satisfied are you with your life as a whole?” Responses ranged from 1 = very satisfied to 5 = very dissatisfied and were reverse scored so that higher scores represent greater life satisfaction [16].

Happiness was assessed with a four-item Subjective Happiness Scale (SHS) [17], with higher scores (4–20) depicting greater levels of subjective happiness. Cronbach’s alpha for the SHS was 0.94 in this sample.

Perceived health status was measured with one item, “In general, would you say that your health is … 1 = excellent, 2 = very good, 3 = good, 4 = fair or 5 = poor?” [18]. Responses were reverse scored with higher scores depicting better health.

#### 2.2.3. Confounding Variables

Socio-demographic items included age, sex, country income and subjective wealth status.

Social support was sourced from three questions of the Social Support Questionnaire [19] (Cronbach’s alpha was 0.94 for this sample).

Tobacco use was measured with the item, “Do you currently use one or more of the following tobacco products (cigarettes, snuff, chewing tobacco, cigars, etc.)?” (“yes” or “no”) [20]. 

Binge alcohol use (past month) was measured with the item, “How often do you have (for men) five or more and (for women) four or more drinks on one occasion?” [21]. 

Body mass index (BMI) was assessed “with standard anthropometric measurements” [22].

### 2.3. Data Analysis

Statistical analysis was performed with STATA software version 14.0 (Stata Corporation, College Station, TX, USA). Parametric statistical procedures were used to show differences in proportions. Multivariable linear regression was used to calculate coefficient estimates and confidence intervals of associations of SB and PA with life satisfaction, happiness and perceived health status. In addition, linear regression was used to estimate the combined relationship of SB and PA with life satisfaction, happiness and perceived health status. Multivariable models were adjusted for relevant confounders, including sex, wealth status, country, social support, tobacco use, binge drinking and body mass index. For the combined linear regression analysis, “the sample was sub-divided based on sedentary and physical activity levels into four groups: (1) high sedentary time (≥8 h) plus low physical activity group (reference category), (2) high sedentary time (≥8 h) plus moderate or high physical activity group, (3) low or moderate sedentary time (<8 h) plus low physical activity group, and (4) low or moderate sedentary time (<8 h) plus moderate or high physical activity group” [22]. Missing data were excluded from the analysis. A *p* < 0.05 was considered significant. 

## 3. Results

### 3.1. Sample Characteristics

Study participants engaged in <4 h (23.5%), 4 to <8 h (42.5%) and ≥8 h (34.0%) SB per day. The PA levels of the participants were as follows: 42.2% low, 36.5% moderate and 21.2% high PA. The mean life satisfaction was 3.14 (range 1–5), happiness 13.08 (range 4–20) and self-rated health status 2.98 (range 1–5). Older students had lower life satisfaction and self-rated health scores than younger students. Happiness scores were higher in students with higher wealth status, living in a country with higher income (middle-income country) (see Table 1).

### 3.2. Associations of SB and PA with Well-Being Indicators

In the final adjusted linear regression model, higher SB (4 to <8 h and ≥8 h) was associated with lower life satisfaction and lower happiness, while higher SB (≥8 h) was associated with lower perceived health, and moderate SB (4 to <8 h) was associated with higher perceived health. In addition, moderate and/or high PA was associated with higher life satisfaction, greater happiness and better perceived health (see Table 2). 

### 3.3. Combined Associations of SB and PA with Well-Being Indicators 

Compared to participants with high SB (≥8 h) and low PA, students who had less than 8 h SB and engaged in moderate or high PA had a significantly greater life satisfaction and better perceived health status, after adjusting for relevant confounders. In unadjusted analysis, students who engaged <8 h in SB and in moderate or high PA had greater happiness (see Table 3). 

## 4. Discussion

Analysis of survey data from a large sample of undergraduate university students from 24 low- and middle-income countries showed a significant independent association between higher SB and moderate and/or high PA with higher life satisfaction, happiness and perceived health. These associations remained statistically significant after controlling for relevant confounders. These findings are consisted with results from studies among university students and other populations in high-income countries [3,4,5,8]. One finding was contrary to expectation, namely that moderate SB (4 to <8 h) was associated with higher perceived health. This result will need further investigation.

Finally, the findings seem to confirm previous results from studies of university students in the USA [6], adolescents in Iceland [5] and school children and adolescents in Iran [23]. This suggests a combined association of SB and PA with life satisfaction, perceived health status and in unadjusted analysis with happiness. Furthermore, longitudinal research is needed to confirm findings from this study.

### Study Limitations

Since this was a cross-sectional investigation, we cannot draw causal conclusions. Results from this study only included university students and cannot be generalised to the general population. In this study, SB and PA were assessed by self-reporting. In the future, objective measures should also be included. The assessment of two of the outcome variables, life satisfaction and perceived health status, were only assessed with single items, and future studies may include more comprehensive measures. 

## 5. Conclusions

The current study showed an independent association of low SB and moderate or high PA with life satisfaction, happiness and perceived health status, in addition to a combined association of low SB and moderate or high PA with life satisfaction and perceived health status. Reducing SB and increasing PA may promote life satisfaction, happiness and perceived health status in this university student population.

## Figures and Tables

**Table 1 ijerph-16-02084-t001:** Sample characteristics and well-being indicators.

Variable	Total	Life Satisfaction	Happiness	Perceived Health Status
Range		1–5	4–20	1–5
	N (%) or M (SD)	M (SD)	M (SD)	M (SD)
All	12,492	3.14 (1.2)	13.08 (2.7)	2.98 (1.0)
Age in years				
18–19	4262 (34.1)	3.22 (1.2)	13.09 (2.7)	3.12 (1.0)
20–21	4393 (35.2)	3.21 (1.2)	13.07 (2.8)	2.94 (1.0)
22–30	3837 (30.7)	2.97 (1.3) *	13.09 (2.7)	2.88 (1.0) *
Sex				
Female	7111 (57.1)	3.12 (1.2)	13.03 (2.6)	2.98 (0.9)
Male	5353 (42.9)	3.18 (1.3)	13.15 (2.8)	2.98 (1.0)
Wealth status				
Low	6910 (55.5)	3.15 (1.2)	12.81 (2.7)	2.97 (1.0)
High	5548 (44.5)	3.14 (1.3)	13.40 (2.7) *	2.99 (1.0)
Country income				
Low/lower middle-income	6322 (50.6)	2.95 (1.2)	12.74 (2.8)	3.01 (1.0)
Upper middle-income	6170 (49.4)	3.31 (1.2) *	13.36 (2.7) *	2.95 (1.0)
Social support				
Low	4320 (34.6)	3.13 (1.2)	12.56 (3.0)	3.00 (1.0)
High	8162 (65.4)	3.15 (1.3)	13.34 (2.5) *	3.01 (1.0)
Current tobacco use				
No	11298 (91.1)	3.17 (1.2)	13.05 (2.7)	2.99 (1.0)
Yes	1097 (8.9)	2.95 (1.3) *	13.19 (2.9)	2.97 (1.0)
Past month binge drinking				
No	11131 (89.1)	3.19 (1.2)	13.06 (2.7)	3.01 (1.0)
Yes	1361 (10.9)	2.77 (1.2) *	13.21 (2.7)	2.77 (1.0) *
Body mass index	21.9 (4.5)			
Sedentary behaviour				
<4 h	2935 (23.5)	3.37 (1.2)	13.25 (2.9)	2.98 (1.0)
4 to <8 h	5315 (42.5)	3.18 (1.2)	13.03 (2.7)	3.04 (1.0)
≥8 h	4242 (34.0)	2.95 (1.2) *	13.03 (2.6) *	2.91 (1.0) *
Physical activity				
Low	5272 (42.2)	3.09 (1.2)	13.04 (2.7)	2.91 (0.9)
Moderate	4573 (36.6)	3.28 (1.2)	13.03 (2.7)	3.06 (1.0)
High	2647 (21.2)	3.02 (1.3) *	13.23 (2.8) *	2.98 (1.1) *

* *p* < 0.001.

**Table 2 ijerph-16-02084-t002:** Coefficient estimates for life satisfaction, happiness and perceived health status.

Variable	Unadjusted Coefficient Estimates: β (95% CI)	Adjusted Coefficient Estimates ^1^: β (95% CI)
Life satisfaction		
SB a day		
<4 h	Reference	Reference
4 to <8 h	−0.13 (−0.19, −0.08) ***	−0.10 (−0.16, −0.04) ***
≥8 h	−0.36 (−0.42, −0.31) ***	−0.21 (−0.27, −0.14) ***
Happiness		
SB a day		
<4 h	Reference	Reference
4 to <8 h	−0.25 (−0.37, −0.12) ***	−0.39 (−0.53, −0.26) ***
≥8 h	−0.27 (−0.40, −0.14) ***	−0.31 (−0.46, −0.17) ***
Perceived health status		
SB a day		
<4 h	Reference	Reference
4 to <8 h	0.06 (0.01, 0.10) **	0.06 (0.02, 0.11) *
≥8 h	−0.08 (−0.13, −0.04) ***	−0.08 (−0.13, −0.03) **
Life satisfaction		
PA		
Low	Reference	Reference
Moderate	0.18 (0.13, 0.23) ***	0.10 (0.04, 0.16) ***
High	−0.09 (−0.15, −0.03) **	0.01 (−0.06, 0.06)
Happiness		
PA		
Low	Reference	Reference
Moderate	0.09 (−0.04, 0.22)	0.00 (−0.11, 0.11)
High	0.35 (0.21, 0.49) ***	0.27 (0.15, 0.39) ***
Perceived health status		
PA		
Low	Reference	Reference
Moderate	0.13 (0.09, 0.17) ***	0.12 (0.08, 0.15) ***
High	0.08 (0.03, 0.12) ***	0.05 (0.01, 0.09) *

SB = sedentary behaviour; PA = physical activity; CI = confidence interval; ^1^ Adjusted for age, sex, wealth status, country, social support, tobacco use, binge drinking, and body mass index; *** *p* < 0.001; ** *p* < 0.01; * *p* < 0.05.

**Table 3 ijerph-16-02084-t003:** Combined associations of sedentary behaviour and physical activity with life satisfaction, happiness and perceived health status.

Variable	*n*	M (SD)	Unadjusted Coefficient Estimates: β (95% CI)	Adjusted Coefficient Estimates ^1^: β (95% CI)
Life satisfaction				
High SB & low PA	1153	2.93 (1.2)	Reference	Reference
High SB & moderate or high PA	2688	2.95 (1.3)	−0.09 (−0.17, −0.02) *	−0.04 (−0.12, 0.05)
Low or moderate SB & low PA	3148	3.18 (1.2)	0.08 (0.01, 0.14) *	−0.02 (−0.09, 0.05)
Low or moderate SB & moderate or high PA	5121	3.28 (1.2)	0.22 (0.16, 0.28) ***	0.10 (0.03, 0.17) **
Happiness				
High SB & low PA	1033	12.94 (2.5)	Reference	Reference
High SB & moderate or high PA	2472	13.06 (2.7)	0.14 (−0.04, 0.31)	0.13 (−0.07, 0.32)
Low or moderate SB & low PA	2741	13.02 (2.8)	0.03 (−0.12, 0.18)	0.07 (−0.09, 0.23)
Low or moderate SB & moderate or high PA	4600	13.19 (2.7)	0.26 (0.12, 0.41) ***	0.13 (−0.03, 0.29)
Self-rated health status				
High SB & low PA	1368	2.88 (0.9)	Reference	Reference
High SB & moderate or high PA	2873	2.93 (1.0)	0.01 (−0.05, 0.07)	−0.04 (−0.12, 0.05)
Low or moderate SB & low PA	3568	2.94 (1.0)	−0.01 (−0.06, 0.04)	−0.02 (−0.09, 0.05)
Low or moderate SB & moderate or high PA	5408	3.07 (1.0)	0.15 (0.10, 0.20) ***	0.13 (0.07, 0.18) ***

SB = sedentary behaviour; PA = physical activity; CI = confidence interval; ^1^ Adjusted for age, sex, wealth status, country, social support, tobacco use, binge drinking, and body mass index; *** *p* < 0.001; ** *p* < 0.01; * *p* < 0.05.

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
