# Peer review of "Sedentary Behaviour, Physical Activity and Life Satisfaction, Happiness and Perceived Health Status in University Students from 24 Countries"

_ijerph, 2019, doi:10.3390/ijerph16122084_

Round 1

Reviewer 1 Report

The manuscript presents data from 12492 university students from 24 countries on aspects of sedentary behaviour, physical activity and wellbeing (life satisfaction, health and happiness). This manuscript represents a very impressive sample size, however more could be done to demonstrate the significance and addition to the literature which this paper makes.

- Clarification is required in the abstract about 4-< 8 – it may just be the formatting but it would be clearer if it was presented as “between 4 and <8” or “4 to <8”. Please also include beta values and significance levels.

- I appreciate that this is a brief report and the authors are adhering to a strict word limit, however, I question whether this is the right format, given the extent of the data. The literature review presents a list of studies in two paragraphs. It would benefit from a more integrated and critical approach to the literature as it is very repetitive and does not make a strong case for the study as it is – it purely just reads like a list of studies.

- You say that “less studies” have looked at the combined impact of PA and SB. This suggests there are some studies – these studies should be discussed in the literature review in order to improve the rationale. I would also encourage you to clearly present the aims/research questions at the end of the literature review.

- For the method, it would be useful to include reliability information for all outcomes.

- While the authors state that methods are described else where, rather than quoting that paper, it would be better to paraphrase so that a clear description is given.

- It is not clear how “standard anthropometric measurements” translate to a self-report questionnaire. Please clarify this in the manuscript. By reading the original paper, it is suggested that participants were actually measured and were not self-reporting. Please make this clearer in the manuscript.

- In the results, it would be useful to include the range for variables, especially BMI.

- In 4.2, the explanation could be clearer of associations between activity and life satisfaction – moderate activity was associated with greater life satisfaction but high activity was associated with lower LS than low activity. This is important and should not be disregarded.

- I would be interested to see whether associations vary across country/region included. This would be valuable analysis to include in a paper which may then have an impact on the discussion.

Author Response

Reviewer I

The manuscript presents data from 12492 university students from 24 countries on aspects of sedentary behaviour, physical activity and wellbeing (life satisfaction, health and happiness). This manuscript represents a very impressive sample size, however more could be done to demonstrate the significance and addition to the literature which this paper makes.

- Clarification is required in the abstract about 4-< 8 – it may just be the formatting but it would be clearer if it was presented as “between 4 and <8” or “4 to <8”. Please also include beta values and significance levels.

Response: Corrected and added

- I appreciate that this is a brief report and the authors are adhering to a strict word limit, however, I question whether this is the right format, given the extent of the data. The literature review presents a list of studies in two paragraphs. It would benefit from a more integrated and critical approach to the literature as it is very repetitive and does not make a strong case for the study as it is – it purely just reads like a list of studies.

Response: This should be sufficient for a brief report.

- You say that “less studies” have looked at the combined impact of PA and SB. This suggests there are some studies – these studies should be discussed in the literature review in order to improve the rationale. 

Response: these studies are presented in the introduction

I would also encourage you to clearly present the aims/research questions at the end of the literature review.

Response: added

- For the method, it would be useful to include reliability information for all outcomes.

Response: there are 3 outcome measures, 2 are single indices, and the happiness one reliability information is provided

- While the authors state that methods are described else where, rather than quoting that paper, it would be better to paraphrase so that a clear description is given.

Response: below is added

Briefly, one university per country was selected by purposeful sampling. In each university, a stratified random sample procedure was used to randomly select undergraduate students for a self-administered questionnaire survey and anthropometric measurements.

- It is not clear how “standard anthropometric measurements” translate to a self-report questionnaire. Please clarify this in the manuscript. By reading the original paper, it is suggested that participants were actually measured and were not self-reporting. Please make this clearer in the manuscript.

Response: this is clarified in above response

- In the results, it would be useful to include the range for variables, especially BMI.

Response: the range for outcome variables is added

- In 4.2, the explanation could be clearer of associations between activity and life satisfaction – moderate activity was associated with greater life satisfaction but high activity was associated with lower LS than low activity. This is important and should not be disregarded.

Response: Corrected

- I would be interested to see whether associations vary across country/region included. This would be valuable analysis to include in a paper which may then have an impact on the discussion.

Response: Country samples were not representative, so it would be difficult to compare.

Reviewer 2 Report

Dear Editor,

I appreciate the opportunity to review this interesting paper entitle "Sedentary behaviour, physical activity and life satisfaction, happiness and perceived health status in university students from 24 countries". In this work the authors explored the associations between sedentary behavior and physical activity with well-being indicators in adults from low- and middle-income countries. It is a well written study and with a very large sample of numerous countries. Clearly a lot of work went into its construction. However, there are limitations and some comments that have to be taken into account. Below are my comments for the authors.

Major comments:

Page 1, lines      27-30: The information is inaccurate. I recommend rephrasing this      paragraph.

Pages 1 and 2,      lines 31-47: What is the main novelty of the study? What does it add new?      Introduction should be more specific.

Page 2, lines 50-54: This information      about the sample is unclear. Moreover, the aim of the study was explored      the associations between sedentary behavior and physical activity with      well-being indicators in adults from low- and middle-income countries and      the authors include participants from high-income countries.

Page 2, lines 56-80: Please provide the      psychometric properties of the instruments used.

Page 2, lines 57-59: The main limitation      of this work is the use of self-reported measures.

Page 2, line 58: The use of      questionnaire for assess sedentary behaviour there is one advantage over      objectively measures such as accelerometry: allow to know the sedentary      patterns and differentiate sedentary modalities (e.g. screen time; social      sedentary time, educational sedentary time...) in the analyzes. In recent      years, studies have shown that the association between sedentary behaviour      and well-being indicators may differ depending on the sedentary modality.

Page 3, lines 90-94: Some      international institutions recommended limit daily time of screen time and      30 minutes of moderate-to-vigorous physical activity per day. However, in      my knowledge, there are no recommendations for total sedentary behaviour      with a scientific basis. It is true that a previous study used as a cut-off      point 3 or 4 hours. Do you have more reasons to use these criteria? How do      you think the use of other cutting points would influence?

Page 6,      lines 124-135: The      paper could be greatly strengthened by a more in-depth discussion.

Minor comments:

Page 1, line 15: Please include      the sample size in the abstract.

Page 1, lines 23 and 24: Please      avoid keywords that have been previously included in the title.

Page 1, line 38: Please remove      the colon previous reference.

Page 2, lines 48-56: I      recommend unifying the sections number 2 (methods) and 3 (measures).

Page 6, lines 147-167: Was this      information omitted for anonymous review?

Pages 6-8, lines 169-237:      Revise the reference list. There are some mistakes in it.

Best regards.

Author Response

Reviewer II

I appreciate the opportunity to review this interesting paper entitle "Sedentary behaviour, physical activity and life satisfaction, happiness and perceived health status in university students from 24 countries". In this work the authors explored the associations between sedentary behavior and physical activity with well-being indicators in adults from low- and middle-income countries. It is a well written study and with a very large sample of numerous countries. Clearly a lot of work went into its construction. However, there are limitations and some comments that have to be taken into account. Below are my comments for the authors.

Major comments:

Page 1, lines      27-30: The information is inaccurate. I recommend rephrasing this      paragraph.

Response: Not sure what is inaccurate here.

Pages 1 and 2,      lines 31-47: What is the main novelty of the study? What does it add new?      Introduction should be more specific.

Response: below is added

Less studies have investigated the independent and combined correlations of SB and PA with life satisfaction, happiness and perceived health in low- and middle-income countries. It would be important to investigate if associations between SB and PA with life satisfaction, happiness and perceived health can be found in a study across culturally different countries and regions in the world. The aim of this investigation was to estimate the independent and combined associations of SB and PA with life satisfaction, happiness and perceived health in university students in 24 low-and middle-income countries.

Page 2, lines 50-54: This information      about the sample is unclear. Moreover, the aim of the study was explored      the associations between sedentary behavior and physical activity with      well-being indicators in adults from low- and middle-income countries and      the authors include participants from high-income countries.

Response: the aim says “university students”, there are no participants from high-income countries

Page 2, lines 56-80: Please provide the      psychometric properties of the instruments used.

Response: They are provided, however, not for single indices

Page 2, lines 57-59: The main limitation      of this work is the use of self-reported measures.

Response: This is in the case of SB and PA acknowledged under study limitations

Page 2, line 58: The use of      questionnaire for assess sedentary behaviour there is one advantage over      objectively measures such as accelerometry: allow to know the sedentary      patterns and differentiate sedentary modalities (e.g. screen time; social      sedentary time, educational sedentary time...) in the analyzes. In recent      years, studies have shown that the association between sedentary behaviour      and well-being indicators may differ depending on the sedentary modality.

Response: Yes, but our SB and PA measure did not assess the different domains

Page 3, lines 90-94: Some      international institutions recommended limit daily time of screen time and      30 minutes of moderate-to-vigorous physical activity per day. However, in      my knowledge, there are no recommendations for total sedentary behaviour      with a scientific basis. It is true that a previous study used as a cut-off      point 3 or 4 hours. Do you have more reasons to use these criteria? How do      you think the use of other cutting points would influence?

Response: the 8 hour SB cut point is referenced 

Page 6,      lines 124-135: The      paper could be greatly strengthened by a more in-depth discussion.

Response: more is added; however, this is a brief report 

Minor comments:

Page 1, line 15: Please include      the sample size in the abstract.

Response: added

Page 1, lines 23 and 24: Please      avoid keywords that have been previously included in the title.

Response: corrected

Page 1, line 38: Please remove      the colon previous reference.

Response: corrected

Page 2, lines 48-56: I      recommend unifying the sections number 2 (methods) and 3 (measures).

Response: corrected

Page 6, lines 147-167: Was this      information omitted for anonymous review?

Pages 6-8, lines 169-237:      Revise the reference list. There are some mistakes in it.

Response: corrected

Reviewer 3 Report

Thanks for the opportunity of providing feedback on this interesting piece of work. I am impressed with the sample. My suggestions are:

1. Abstract

1.1. Include the sample size

2. Introduction

2.1. Correlation and association seem to be interchangeably used. I would use "association" as a generic term.

2.2. "Island" must be "Iceland".

2.3. Instead of mentioning so many individual studies, references to reviews may be of interest. In this context, the findings of a recent review in children and adolescents may be relevant (https://www.ncbi.nlm.nih.gov/pubmed/30993594).

2.4. The rationale and importance of conducting this study are not outlined. Please, note that a lack of research (as indicated in the last paragraph) is not a strong argument. 

2.5. I would strongly suggest introducing the aim at the end of the introduction by following the structure "Therefore, the aim of this investigation was....".

3. Methods

3.1. Sample and procedure. Please, list the countries in which the study was performed as well as the review boards and codes of ethical approval of each country.

3.2. Exposure variables. Physical activity was classified as <4, 4 to 8 and > 8 hours while sedentary behaviour was classified as low, moderate, and high. First, use times or labels for both variables. Second, provide the rationale for using these cut-off values for the categories and show such cut-off values for sedentary behaviour. Third, I would say that this work measured "sedentary time" instead of "sedetary behaviours" as it looks at time instead of behaviours (e.g., watching TV versus reading a book). 

3.3. Perceived health. I miss what are the numeric values for each possible answer.

3.4. Body mass index. What does "with standard anthropometric measurements" mean?

3.5. Analyses. Why was the time spent in physical activity and sedentary behaviours classified in categories instead of using their continuous values? This is probably the most important methodological limitation of the paper. 

3.6. Other issues: How were missing data managed? A flow chart of participants would be appreciated. 

4. Results. The decision of not using continuous data of PA and SB negatively impact on the results. For instance, table 3 would be much more informative by computing the interaction term of continuous PA and SB. 

5. Discussion. Too brief. For instance, there is no discussion on why participants with moderate SB reported worse happiness and perceived health than those with >8 h of SB (see, Table 2). Also, why participants with moderate PA reported better life satisfaction and perceived health (but not happiness) than those highly active (see, Table 3)?

6. Limitations. First, attention should be paid to the fact that results from students are not generalizable to the general population. Second, in order to support the statement about the importance of objective measurements a reference to this work can be added (https://www.ncbi.nlm.nih.gov/pubmed/27973914). 

Overall, the work is of interest and suits well to the journal scope. I would like to congratulate the authors for their hard work!

Author Response

Reviewer III

anks for the opportunity of providing feedback on this interesting piece of work. I am impressed with the sample. My suggestions are:

1. Abstract

1.1. Include the sample size

Response: added

2. Introduction

2.1. Correlation and association seem to be interchangeably used. I would use "association" as a generic term.

Response: corrected

2.2. "Island" must be "Iceland".

Response: corrected

2.3. Instead of mentioning so many individual studies, references to reviews may be of interest. In this context, the findings of a recent review in children and adolescents may be relevant (https://www.ncbi.nlm.nih.gov/pubmed/30993594).

Response: This review is not included because it deals with children and adolescents, while our study and review focuses on adults

2.4. The rationale and importance of conducting this study are not outlined. Please, note that a lack of research (as indicated in the last paragraph) is not a strong argument. 

2.5. I would strongly suggest introducing the aim at the end of the introduction by following the structure "Therefore, the aim of this investigation was....".

Response: below is added

Less studies have investigated the independent and combined associations of SB and PA with life satisfaction, happiness and perceived health in low- and middle-income countries. It would be important to investigate if associations between SB and PA with life satisfaction, happiness and perceived health can be found in a study across culturally different countries and regions in the world. The aim of this investigation was to estimate the independent and combined associations of SB and PA with life satisfaction, happiness and perceived health in university students in 24 low-and middle-income countries.

3. Methods

3.1. Sample and procedure. Please, list the countries in which the study was performed as well as the review boards and codes of ethical approval of each country.

Response: below is added

(Bangladesh, Cameroon, Columbia, Grenada, India, Ivory Coast, Kenya, Jamaica, Kyrgyzstan, Laos, Madagascar, Malaysia, Mauritius, Myanmar, Namibia, Nigeria, Philippines, Russia, South Africa, Thailand, Tunisia, Turkey, Venezuela, and Vietnam).

Participating students signed informed consent forms and all implementing institutions obtained ethics approvals: Ethics Review Committee North South University, Ethics Committee of the University of Yaoundé, Universidad de Pamplona Ethics Committee, St. George’s University Institutional Review Board, Ethics Committee of Institute of Technology and Institute of Sciences at GITAM (Gandhi Institute of Technology and Management) University, Félix Houphouët Boigny University Ethics Committee, University of Nairobi University of Nairobi Ethics and Research Committee, University of the West Indies Ethics Committee, Kyrgyz State Medical Academy Ethics Committee, The Ethics Committee of the University of Health Sciences, Ethics Committee of the University of Antananarivo, University of Malaya Medical Ethics committee (MECID 201412–905), University of Mauritius Research Ethics Committee, Research and Ethical Committee of University of Medicine 1, Research Ethics Committee of the University of Namibia, Ethics Review Committee Obafemi Awolowo University, Committee of the Western Visayas Health Research, Ethics Committee of the Peoples’ Friendship University of Russia, Medunsa Research and Ethics Committee (MREC/H/275/2012), Committee for Research Ethics (Social Sciences) of  Mahidol University (MU-SSIRB 2015/ 116(B2), National Ethics Committee for Health Research at Institut National de la Santé Publique, Ethics Committee Istanbul University, Ethics Committee of  the Universidad Central de Venezuela, Committee Committee of Research Ethics of Hanoi School of Public Health.

3.2. Exposure variables. Physical activity was classified as <4, 4 to 8 and > 8 hours while sedentary behaviour was classified as low, moderate, and high. First, use times or labels for both variables. 

Response: corrected

Second, provide the rationale for using these cut-off values for the categories and show such cut-off values for sedentary behaviour. 

Response: This is referenced under the SB measure

Third, I would say that this work measured "sedentary time" instead of "sedetary behaviours" as it looks at time instead of behaviours (e.g., watching TV versus reading a book). 

Response: sedentary time or behaviour, sitting is a behaviour

3.3. Perceived health. I miss what are the numeric values for each possible answer.

Response: Corrected, as below

Perceived health status was measured with one item, “In general, would you say that your health is … 1=excellent, 2=very good, 3=good, 4=fair or 5=poor

3.4. Body mass index. What does "with standard anthropometric measurements" mean?

Response: as below

Students were weighed and measured by trained researchers using standardised procedures [48]. Standing height ofeach student was measured to the nearest 0.1 cm without shoes, using a stature meter. Participantswere weighed to the nearest 0.01 kg, in their light clothes, on a load-cell-operated digital scale hav-ing a weighing capacity of 140 kg. The scale usedduring the survey is first calibrated with a standardweight and checked on a daily basis. 

3.5. Analyses. Why was the time spent in physical activity and sedentary behaviours classified in categories instead of using their continuous values? This is probably the most important methodological limitation of the paper.

Response: This was done as in most studies (to better compare) 

3.6. Other issues: How were missing data managed? 

Response: Missing data were excluded from the analysis.

A flow chart of participants would be appreciated. 

Response: more about participants sampling has been added

4. Results. The decision of not using continuous data of PA and SB negatively impact on the results. For instance, table 3 would be much more informative by computing the interaction term of continuous PA and SB. 

Response: This was done as in most studies (to better compare) 

5. Discussion. Too brief. For instance, there is no discussion on why participants with moderate SB reported worse happiness and perceived health than those with >8 h of SB (see, Table 2). Also, why participants with moderate PA reported better life satisfaction and perceived health (but not happiness) than those highly active (see, Table 3)?

Response: more is added; yet, this is a brief report

6. Limitations. First, attention should be paid to the fact that results from students are not generalizable to the general population.

Response: added

 Second, in order to support the statement about the importance of objective measurements a reference to this work can be added (https://www.ncbi.nlm.nih.gov/pubmed/27973914). 

Response: not sure how below can be of help here

The discordance between subjectively and objectively measured physical function in women with fibromyalgia: association with catastrophizing and self-efficacy cognitions. The al-Ándalus project.

Overall, the work is of interest and suits well to the journal scope. I would like to congratulate the authors for their hard work!

Round 2

Reviewer 2 Report

I thank the authors for providing answers to my comments and in general making appropriate changes to their manuscript. I have only some few additional comments:

- Page 1, lines 28-31: The first sentence is not closely related with the second sentence and the aim of this study.

- The information about the sample is unclear around the Table 1. The aim of the study was explored the [...] in adults from low- and middle-income countries but several times authors talk about higher income countries.

Yours sincerely.

Author Response

I thank the authors for providing answers to my comments and in general making appropriate changes to their manuscript. I have only some few additional comments:

- Page 1, lines 28-31: The first sentence is not closely related with the second sentence and the aim of this study.

Response: “Physical inactivity” is added in the first sentence. Further, the first sentence shows the general impact of sedentary behaviour and physical inactivity on physical and mental morbidity, while the second sentence refers to positive health and well-being

- The information about the sample is unclear around the Table 1. The aim of the study was explored the [...] in adults from low- and middle-income countries but several times authors talk about higher income countries.

Response: this is corrected to “ low or lower middle-income”  and “upper middle-income countries”